# Synthesis, Optical, Chemical and Thermal Characterizations of PMMA-PS/CeO_2_ Nanoparticles Thin Film

**DOI:** 10.3390/polym13071158

**Published:** 2021-04-04

**Authors:** Areen A. Bani-Salameh, A. A. Ahmad, A. M. Alsaad, I. A. Qattan, Ihsan A. Aljarrah

**Affiliations:** 1Department of Physical Sciences, Jordan University of Science & Technology, P.O. Box 3030, Irbid 22110, Jordan; areensalameh92@gmail.com (A.A.B.-S.); sema_just@yahoo.com (A.A.A.); ihsanaljarrah@gmail.com (I.A.A.); 2Department of Physics, Khalifa University of Science and Technology, Abu Dhabi P.O. Box 127788, United Arab Emirates; issam.qattan@ku.ac.ae

**Keywords:** hybrid thin films, optical properties, CeO_2_ nanoparticles, polystyrene (PS), polymethyl methacrylate (PMMA), chemical properties, thermal properties

## Abstract

We report the synthesis of hybrid thin films based on polymethyl methacrylate) (PMMA) and polystyrene (PS) doped with 1%, 3%, 5%, and 7% of cerium dioxide nanoparticles (CeO_2_ NPs). The As-prepared thin films of (PMMA-PS) incorporated with CeO_2_ NPs are deposited on a glass substrate. The transmittance T% (λ) and reflectance R% (λ) of PMMA-PS/CeO_2_ NPs thin films are measured at room temperature in the spectral range (250–700) nm. High transmittance of 87% is observed in the low-energy regions. However, transmittance decreases sharply to a vanishing value in the high-energy region. In addition, as the CeO_2_ NPs concentration is increased, a red shift of the absorption edge is clearly observed suggesting a considerable decrease in the band gap energy of PMMA-PS/CeO_2_ NPs thin film. The optical constants (n and k) and related key optical and optoelectronic parameters of PMMA-PS/Ce NPs thin films are reported and interpreted. Furthermore, Tauc and Urbach models are employed to elucidate optical behavior and calculate the band gaps of the as-synthesized nanocomposite thin films. The optical band gap energy of PMMA-PS thin film is found to be 4.03 eV. Optical band gap engineering is found to be possible upon introducing CeO_2_ NPs into PMMA-PS polymeric thin films as demonstrated clearly by the continuous decrease of optical band gap upon increasing CeO_2_ content. Fourier-transform infrared spectroscopy (FTIR) analysis is conducted to identify the major vibrational modes of the nanocomposite. The peak at 541.42 cm^−1^ is assigned to Ce–O and indicates the incorporation of CeO_2_ NPs into the copolymers matrices. There were drastic changes to the width and intensity of the vibrational bands of PMMA-PS upon addition of CeO_2_ NPs. To examine the chemical and thermal stability, thermogravimetric (TGA) thermograms are measured. We found that (PMMA-PVA)/CeO_2_ NPs nanocomposite thin films are thermally stable below 110 °C. Therefore, they could be key candidate materials for a wide range of scaled multifunctional smart optical and optoelectronic devices.

## 1. Introduction

Nanocomposites based on blending polymers with inorganic nanoparticles have attracted much attention owing to their projected extraordinary thermal, optical, electrical and antibacterial properties [1,2,3,4]. The motivation for using inorganic materials stems from their high thermal stability, good electrical properties and high refractive index *n* [5,6]. However, previous studies indicate several drawbacks and insufficient capability of inorganic nanoparticles to serve a variety of modern device applications [7]. Of these disadvantages, there are the deficiency of elasticity, high cost and their high densities. As a result, several researchers were motivated to search for nanocomposite materials based on blending inorganic nanoparticles with organic materials to improve their properties and obtain nanocomposites with better features. Owing to the outstanding properties of organic materials such as good flexibility, lower weight compared to inorganic materials, as well as the easiness of preparation, low processing cost, good influence resistance [8,9,10], recyclability and being environmentally friendly, they serve as excellent candidates for inorganic-organic nanocomposites [11,12].

High transparency and high refractive index are two essential optical parameters that are highly demanded for the manufacturing of smart multi-functional optoelectronics devices [9]. Careful design of such nanocomposites could yield materials that can be employed in sophisticated sensors [13,14], optical crystals [15], micro-lenses for imaging and medical applications [16,17] and ultra-fast data transmission [10]. Polymers with high refractive indices are key candidate materials for smart-scaled multi-functional materials. For instance, poly(thiophene) exhibits a refractive index of 2.12 at a wavelength of 632 nm [16,18,19,20]. However, they are difficult to prepare. Despite the high absorption coefficient and apart from the difficulty of preparing, it absorbs light in the visible light region and has high optical dispersion [15]. In light of the above issues, overcoming these limitations and developing organic materials with better properties becomes of utmost importance.

The main challenge is to explore the possibility of preparing polymers thin films with specific properties for specific applications. Polymethyl methacrylate (PMMA) polymer has an excellent optical, electrical, mechanical, and thermal characteristic. The exceptional properties of PMMA such as, high transparency, environmental stability, low cost, easy preparation and shaping at low temperature make it an excellent candidate for fabricating thin films [21,22,23]. However, its application is limited at higher temperatures owing to its relatively poor thermal stability [24]. To overcome this problem, elaborated technological techniques are applied to improve different characteristics of PMMA polymer [24,25,26]. Polystyrene (PS) is one of the most common thermoplastic polymers. It is colorless and transparent in the visible region. PS has a good formability, a good rigidity, electric and thermal insulation, easy processing and long-term stability [27,28,29]. The outstanding properties of polystyrene such as its low cost and high refractive index of 1.59 at a wavelength of 632 nm make it an excellent choice for several optical applications [8].

Controlling the properties of materials is like sculpturing uniquely on scrolls. The properties of the films can be tuned in several ways, such as changing the film thickness [30,31,32,33,34] or by developing hybrid films of the metal oxide polymer [8,35]. In addition, the properties of the polymers can be improved by introducing nanoparticles into the polymer matrix. This is an approach that opens the door to many applications in various fields. The merging of nanoparticles into the polymer matrix improves their optical, mechanical, thermal and electronic properties [8,36,37]. Due to the simple synthesis and low cost of preparation, the merging of inorganic building blocks in the organic matrix is an effective way to improve the properties while maintaining high transparency [38,39]. Amongst nanoparticles, cerium oxide nanoparticles (CeO_2_ NPs) have attracted much attention for their high stability, surface chemistry, and biocompatibility [40,41,42]. CeO_2_ NPs are transparent in the visible region and have a refractive index of 2.2 at a wavelength of 632 nm [8,43]. Pure CeO_2_ exhibits a wide indirect optical band gap and energy-wide band gap that operates effectively in the ultraviolet (UV) region and thus it could be an excellent choice for different optical and electronic applications [44,45].

Optics are playing a crucial role in many of our day-to-day applications. The refractive index is one of the most significant parameters in photonics. An increase in the efficiency of the photonic devices, like LEDs, modern solar systems, can be performed by engineering the refractive index mismatch of materials used in the optical devices. The novelty of this work is underlined by refractive index modulation using polymer nanocomposites treated with inorganic fillers. Other related optical parameters of the fabricated optical devices can be tuned accordingly. By careful choice of synthetic methods and manipulating the distinctive physics of the polymeric nanocomposites in such materials, novel-scaled functional polymer–inorganic nanocomposites can be designed and manufactured for new and interesting optoelectronic applications.

## 2. Experimental Details and Techniques

Polystyrene (PS) with a molecular weight 104.1 g/mol, polymethyl methacrylate (PMMA) with a molecular weight 3617 g/mol, and Ceria NPs with molecular weight 172 g/mol were purchased from Sigma Aldrich (St. Louis, MO, USA). Polystyrene (PS) solution was prepared in a conical flask by dissolving 2 g of PS in 200 mL of tetrahydrofuran (THF) Sol(A) and then placed on a stirrer for 1 h. Polymethyl methacrylate (PMMA) solution was prepared by dissolving 2 g of PMMA in 200 mL THF Sol(B) under continuous stirring for 1 h. Immediately after that, Sol (A) was added to Sol (B) under continuous stirring to synthesize 1:1 co-polymeric matrix. The polymeric mixture is expected to exhibit extraordinary physical, thermal and optical properties. Cerium oxide nanoparticles purchased from (Sigma Aldrich, St. Louis, MO, USA) of size (25–50) nm were added to as-prepared copolymers matrix with different concentration ratios (1%, 3%, 5% and 7%). To ensure that CeO_2_ nanoparticles were incorporated homogenously into the PMMA-PS matrix, the solution was alternatively mixed on a magnetic stirrer and a sonication rod. The substrates were cleaned and rinsed using ethanol and distilled water. The PMMA-PS/CeO_2_ nanocomposite thin films were synthesized by dip-coating technique. The two layers were dried at 70 °C for 30 min for each layer. The effect of introducing CeO_2_ nanoparticles on the optical properties was performed using UV-Vis spectrophotometer (U-3900H) (Hitachi, Fukuoka, Japan) with a total internal integrating sphere. Particularly, transmittance T% (λ) and reflectance R% (λ) of PMMA-PS/CeO_2_NPs thin films at room temperature in (250–700) nm spectral range are measured and interpreted. Electron Scanning Microscope (SEM) (Quanta FEG 450) (FELMI-ZFE, Graz, Austria) is utilized to investigate the surface morphology of as-prepared thin films. Thermogravimetric analysis (TGA) technique is used to study thermal stability of as-synthesized doped polymeric thin films. To identify the major vibration modes and types of different bonding networks of the PMMA-PS/CeO_2_ nanocomposites, Fourier transform infrared spectroscopy (FTIR) (Bruker Vertex 80 and Hyperion 2000 microscope) (Bruker Optics, Karlsruhe, Germany) analysis is conducted.

## 3. Result and Discussion

### 3.1. Optical Properties of PMMA-PS/Ce NPs Thin Film

#### 3.1.1. Transmittance and Reflectance

UV-Vis spectrophotometer was used to explore and investigate the optical properties of PMMA-PS thin films doped with various Ce NPs concentrations. Figure 1 shows the transmittance of PMMA-PS/Ce NPs thin films. Analysis and interpretation of transmittance data can be partitioned into two spectral regions. Namely, low energy region (700≥λ≥350) nm in which all samples exhibit a high transparency of about 87%. The high energy region (250≤λ≤350) nm is where transmittance starts to decay to a vanishing value. This region contains the absorption edge [46] that is red-shifted upon increasing the concentration of Ce NPs in the polymeric matrix suggesting a significant reduction of optical band gap energy of (PMMA-PS)/CeO_2_ nanocomposite thin films [47]. The incorporation of CeO_2_ NPs into polymeric matrix leads to a compression of the host matrix. Compressive strain introduced into PMMA-PS matrix can cause a red shift as a result of the changes of a built-in electric field. The polarization should be affected by doping and causes a red shift. 

Figure 2 shows the reflectance of (PMMA-PS)/CeO_2_ nanocomposite thin films. It can be clearly observed that reflectance of PMMA-PS, PMMA-PS/1%, 3%, 5%, and 7% CeO_2_NPs thin film exhibit values of (7.6–10.1%), (9.2–9.8%), (9.3–10.2%), (8.7–10.4%), (9–11.5%), respectively.

#### 3.1.2. Extinction Coefficient and Refractive Index

The extinction coefficient *k* for all samples was calculated using the formula k=αλ/4π where α is the absorption coefficient defined by α=(1/d)ln(1/T) where *d* is the thickness of thin films estimated to be 250 nm [48,49]. The use of inorganic ceria nanoparticles into the polymer matrix can provide high-performance novel materials that find applications in many industrial fields. With this respect, frequently considered features are optical properties such as light absorption. 

Figure 3 shows the extinction coefficient *k* calculated in the spectral range (250–700) nm as a function of incident wavelength. For the spectral range 700≥λ≥350 nm, *k* exhibits a vanishing value for all investigated thin film samples; this means that thin films allow electromagnetic waves to pass through without any decay or damping for photons with wavelengths λ ≥ 350 nm. In the high frequency regions, 250≤λ≤350 nm, *k* increases and attains a maximum value at 290 nm. This can be attributed to extremely high absorption of the energetic EM waves in this region. Figure 3 indicates that such energetic EM waves having energies very close to the optical band gap energy of the nanocomposite thin film are largely absorbed. Exceptionally, (PMMA-PS)/CeO_2_ 3% nanocomposite thin films exhibit the highest extinction coefficient indicating that energetic EM waves are completely absorbed in this case.

Refractive index (*n*) is generally associated with the electronic polarization of ions and local field inside optical materials. Compared to inorganic solids, optical applications of polymers are often limited due to the relatively narrow range of the refractive index. Thus, the introduction of inorganic nanoparticles into a polymer matrix can result in polymeric nanocomposites with extreme refractive index, which finds potential applications in lenses, optical filters, reflectors, optical waveguides, optical adhesives, solar cells, or antireflection films. To elucidate a deeper insight into optical properties, refractive index (*N*) is essentially composed of real part (*n*) and imaginary part (*k*); (N=n+ik) where $n=(\left(1+R\;)/(1-R\right)+\sqrt{4R\left(1-R^{2}\right)-k^{2}}$ [50]. Figure 4 shows that *n* of PMMA-PS exhibits values ranging between 1.76 and 2.13. Introducing 1% of CeO_2_ NPs into polymeric matrix leads to a slight increase of *n* (1.86–2.15). As the CeO_2_ NPs concentrations is increased to 3%, 5%, and 7%, *n* continuously increases to (1.88–2.15), (1.83–2.18) and (1.85–2.26). Consequently, (PMMA-PS)/CeO_2_ nanocomposite thin films could be potential candidates as excellent reflective material [51]. 

#### 3.1.3. Band Gap Energy Eg

Optical band gap energy $E_{g}$ of as-prepared doped thin films is investigated using Tauc plot model. According to this model, $\left(\alpha hv\right)=B{\left(hv-E_{g}\right)}^{\left(1/2\right)}\;$, where *B* is a constant related to the type of the thin film. Figure 5 shows the relationship between the energy of incident photons (E=hv) and ${\left(\alpha hv\right)}^{2}$. The optical band gap energy *E_g_* of (PMMA-PS)/CeO_2_ nanocomposite thin films with various CeO_2_ NPs concentrations is obtained by extrapolating the liner part of Tauc plot to the interception of the incident photon energy (hv). The obtained optical band gap energy of PMMA-PS is calculated to be *E_g_* = 4.03 eV consistent with previously reported values [52,53]. As the CeO_2_ NPs concentrations is increased to 1%, 3%, 5%, and 7%, optical band gap decreases to 3.97 eV, 3.76 eV, 3.63 eV, and 3.6 eV respectively. Thus, band gap engineering could be achieved effectively by inserting a specific concentration of CeO_2_ NPs in the polymeric thin films. 

Refractive index dispersion is one of the most crucial parameters. Moreover, calculating dispersion energies is essential to obtain a deeper insight into the applications of (PMMA-PS)/CeO_2_ nanocomposite thin films for optical devices [51]. Therefore, refractive index and dispersion energies must be studied carefully to specify the potential application of the material [54]. 

Figure 6 shows optical band gap energy *E_g_* plotted versus CeO_2_ NPs concentration (%). It can be noticed that *E_g_* decreases exponentially with ceria NPs concentration.

##### Wemple DiDomenico Model

Wemple DiDomenico model (WDD) is a classical single effective-oscillator model. This model can be utilized to calculate key optical dispersion parameters such as effective single oscillator energy $E_{o}$ and dispersion energy $E_{d}$ for certain optical materials. The $E_{o}$ parameter provides essential information about the band structure of the polymeric thin film, while $E_{d}$ is associated with the mean potency of interband photosensitive transitions and the structural fluctuations [55]. Furthermore, WDD model can be employed to estimate other optical parameters such as the zero frequency-refractive index $\left(n_{0}\right)$, zero-frequency dielectric constant $\varepsilon_{0}$, and the spectral moments $M_{-1}$ and $M_{-3}$. The relationships of different parameters can be expressed as,
(1)$$\;\;\;\;\;\;\;\;\left(n^{2}-1\right)\;=\frac{E_{d}\;E_{0}}{E_{0}^{2}-hv^{2}}$$

Equation (1) can be rewritten as,
(2)$${\left(n^{2}-1\right)}^{-1}\;=\;\frac{E_{0}}{E_{d}}-\;\;\frac{hv^{2}}{E_{0}E_{d}}$$

The energy of incident photon (hv) can be plotted against ${\left(n^{2}-1\right)}^{-1}$ to obtain the values of dispersion parameters from the slope of the obtained straight line ($E_{d}E_{o}{)}^{-1}$ and the interception with the vertical axis $\left(E_{0}/E_{d}\right)$. Figure 7a,b display ${\left(n^{2}-1\right)}^{-1}$ versus ${\left(hv\right)}^{2}$ and ${\left(n^{2}-1\right)}^{-1}$ versus ${\left(\lambda \right)}^{-2}$ of (PMMA-PS)/CeO_2_ nanocomposite thin films incorporated with various concentrations of CeO_2_ NPs. The values of the two important dispersion parameters $E_{d}$ and $E_{o}$ obtained from the two plots are listed in Table 1. Careful inspection of the values of $E_{d}$ and $E_{o}$ parameters of PMMA-PS are found to be 6.893 eV and 4.063 eV, respectively. The corresponding attained values of (PMMA-PS)/CeO_2_ 1% are 14.484 eV and 6.333 eV, indicating a significant increase as only a small concentration of CeO_2_ NPs are incorporated into polymeric films. 

By rewriting Equation (2) and setting hv=0 [56], the $\varepsilon_{0}$ and $\left(n_{0}\right)$ are related by the formula,
(3)$$\varepsilon_{0}=n_{0}^{2}=1+\frac{E_{d}}{E_{0}}$$

The obtained values of $\varepsilon_{0}$ are presented in Table 1. The calculated values of $n_{0}$ are consistent with the theoretical and the experimental values of the normal refractive index. The estimated values of $\varepsilon_{0}$ and $n_{0}$ of PMMA-PS are found to be 2.696 and 1.642, respectively. Introducing CeO_2_ NPs into PMMA-PS boost the values of $\varepsilon_{0}$ and $n_{0}$ for all investigated thin film samples. In addition, the interplay between $E_{d},$
$E_{o}$ and the optical oscillator strength (*f*) of the optical transition between the initial state and the final state can be expressed as $f=E_{d}\;E_{o}$ [57,58]. The obtained values of *f* are displayed in Table 1. Moreover, the effective single oscillator moments $M_{-1}$ and $M_{-3}$ can be correlated with dispersion parameters [59,60,61].
(4)$$E_{0}^{2}=\frac{M_{-1}}{M_{-3}}$$
(5)$$E_{d}^{2}=\frac{M_{-3}^{3}}{M_{-1}}$$

As demonstrated by Table 1, the values of optical moments $M_{-1}$ and $M_{-3}$ of PMMA-PS are obtained to be 1.696 and 0.102 (eV^−2^), respectively; while for (PMMA-PS)/1% CeO_2_ NPs, the value of $M_{-1}$ increases to 2.286 and the value of $M_{-3}$ decreases to 0.056 eV^−2^. This behavior could be explained in terms of the significant decrease in the polarization of (PMMA-PS)/CeO_2_ NPs nanocomposite thin films [62].

##### Sellmeier Oscillator Parameters

From another perspective, we employ Sellmeier oscillator model to elucidate the dispersion in thin films in terms of the average oscillator wavelength ($\lambda_{0}$) and on the oscillator length strength ($S_{0}$). Refractive index and the squared wavelength at higher wavelength are related to $\lambda_{0}$ and $S_{0}$ by the formula,
(6)$$n^{2}-1=\frac{\left(S_{0}\lambda_{0}\right)}{1-(\lambda_{0}/\lambda^{2})\;}$$

By plotting ${\left(n^{2}-1\right)}^{-1}$ versus $\lambda^{-2}$, we calculate $S_{0}$ from the slope of the resulting straight line $\left(1/S_{0}\right)$. The values of ($\lambda_{0}$) can be obtained from the intercept with the vertical axis $\left(1/\lambda_{0}^{2}S_{0}\right)$ as can be clearly seen from Figure 7b. It clearly shows that the refractive index at higher wavelength adopts Sellmeier’s dispersion relation. The calculated values of $S_{0}$ and $\lambda_{0}$ are tabulated in Table 1. It reveals that $S_{0}$ and $\lambda_{0}$ of PMMA-PS are found to be 306.200 × 10^−5^ and 306.200 nm, respectively. For PMMA-PS/CeO_2_ NPs nanocomposites *S*_0_ decreases and *λ*_0_ increases. These tendencies hold up to 5% of CeO_2_ NPs incorporated into polymeric films.

##### Urbach Energy

To obtain a deeper insight into optical properties of thin films, order of crystallinity for PMMA-PS/CeO_2_ nanocomposite thin films is investigated by calculating Urbach energy *E_U_*. The absorption coefficient is related to *E_U_* via $\alpha =\alpha_{0}\exp\left(hv/E_{U}\right)$, where $\alpha_{0}$ is a constant. By plotting ln(α) versus incident photon energy (hv), $E_{U}$ can be determined by extrapolating the straight line below the absorption band edge. The estimated values of $E_{U}$ of PMMA-PS/CeO_2_ nanocomposite samples are presented in Table 1. For unloaded PMMA-PS thin film, $E_{U}$ is found to be 182.495 meV. It is observed that value of *E_U_* of PMMA-PS/7% CeO_2_ has increased to 207.675 meV suggesting a significant disorder and surface interactions in the polymeric thin films loaded with ceria nanoparticles.

### 3.2. FTIR Analysis

Fourier Transform Infrared Spectroscopy (FTIR) is employed to explore and identify the vibrational bands of the loaded PMMA-PS thin films. Figure 8 shows the FTIR spectra of PMMA-PS and PMMA-PS doped by CeO_2_ NPs. The vibrational bands observed in the FTIR spectrum are typical of those of PMMA and PS polymers. The vibrational bands associated with bending of C–H bonds are registered in the 1000–700 cm^−1^ spectral range. The vibrational bands located in the 1000–1300 cm^−1^ range are assigned to C–O stretching. The vibrational bands recorded in between 1300–1400 cm^−1^ are assigned to –CH_3_ bending, while a band at 1449 cm^−1^ could be ascribed to the –CH_2_ bending. Band at 1484 cm^−1^ could be ascribed to the C=C bonds. The bands appearing between 1600–1800 cm^−1^ are associated with C=O bonds. Bands identified between 2800–3200 cm^−1^ are allocated to the C–H stretching. The six IR bands located in the 1000–1300 cm^−1^ spectral range are associated with C–O vibrational modes. As the dipole moment changes due to the vibrations of atoms, two IR bands are associated with symmetric stretch, two with asymmetric stretch and two with the C–O bending. The wide spectral range identifies the locations of different IR bands in the nanocomposite thin films. Finally, the peak appearing at 541.42 cm^−1^ in the nanocomposites is clearly associated with Ce–O suggesting homogenous incorporation of CeO_2_ NPs into the co-polymer’s matrices. Furthermore, significant changes observed in width and intensity of the vibrational bands of PMMA-PS upon addition of CeO_2_ NPs indicate the strong influence of ceria NPs on the spectroscopy of the blended polymer. Table 2 presents the peak positions of all major vibrational bands of PMMA-PS doped by CeO_2_ NPs. The two main factors that influence the intensity of an IR absorption band are the intermolecular bonding between PMMA-PS matrix and Ce NPs as well as the change in dipole moment that occurs during a vibration.

### 3.3. Thermogravimetric Analysis (TGA)

To elucidate thermal stability of doped polymeric thin films investigated in this study, thermogravimetric analysis (TGA) (weight loss %) with respect to temperature and derivative thermogravimetric analysis (DTG) are measured for PMMA-PS, and PMMA-PS incorporated CeO_2_ NPs. A weight loss and differential thermogravimetry curve (first derivative of the weight with respect to temperature) of PMMA-PS and PMMA-PS incorporated CeO_2_ NPs are shown in Figure 9. Two main regions of weight loss and first derivative weight loss are observed. The first is identified in the 100–200 °C range. In this region, the estimated weight loss is estimated to be 3–12% that could be attributed to adsorbed water. In the second region, mainly observed between 300 °C and 400 °C, the weight loss decreases from 90% to 20%. Such a large weight loss is clear indication of thermal decomposition suggesting that doped polymeric thin films exhibit low chemical stability at high temperature. Furthermore, a maximum rate of weight loss is observed at 330 °C, for PMMA-PS thin films. This maximum rate shifts slightly towards a high temperature region as the concentration of CeO_2_ NPs inserted into PMMA-PS matrix is increased.

Figure 10 and Figure 11 show the FTIR spectra of PMMA-PS and PMMA-PS/5%CeO_2_ for thin film samples processed at different annealing temperatures. For thin film samples annealed at temperatures 200–400 °C, absorbance of both PMMA-PS and PMMA-PS/CeO_2_ exhibits a sharp shrink away. This could be attributed to the elimination of C–H bending of PMMA-PS, and C–H bending and C–O stretching of PMMA-PS/5% CeO_2_.

### 3.4. Surface Morphology of PMMA-PS/CeO_2_ Thin Films

The ability to control the orientation of block copolymer thin film features relative to the surface is key to the material’s usefulness for patterning. For example, a surface appears as meandering fingerprint line/space patterns for CeO_2_ to be homogenously inserted and distributed. For PMMA-PS, several well-established and effective surface pre-treatments for dictating the domain orientations of self-assembled patterns are essentially possible. Even with appropriate surface pre-treatments, self-assembly of PMMA-PS patterns are relatively sensitive to the concentration of CeO_2_ NPs. Exciting recent advances in block copolymer-based patterning have used self-assembly to achieve alignment and registration of features by directing meandering self-assembled fingerprint patterns. Surface morphology of (PMMA-PS)/CeO_2_ NPs at 20 μm magnification are presented in Figure 12. Figure 12a shows that undoped PMMA-PS thin films exhibit an organized texture. Figure 12b–d show a small effect of the CeO_2_ NPs immersed into thin film matrix on the surface of doped PMMA-PS nanocomposite thin films. Furthermore, we examine SEM micrographs to investigate the morphology and dispersion of CeO_2_ NPs on the surface of PMMA-PS films. Good dispersion of CeO_2_ NPs on the surface of the PMMA-PS thin films is revealed. This provides a substantial evidence of the validity of our synthesis process of obtaining CeO_2_ NPs. SEM images indicate that measured size of CeO_2_ NPs is in 25–50 nm.

## 4. Conclusions

In summary, (PMMA-PS)/CeO_2_ nanocomposite thin films doped with different concentrations of CeO_2_-NPs (0 to 7%) are synthesized and deposited on glass substrates via dip-coating technique. As-grown thin films are investigated to elucidate the spectral behavior of key optical parameters such as transmittance, reflectance, absorption coefficient, refractive index, and extinction coefficient. Furthermore, a combination of classical models such as Tauc, Wemple DiDomenicol, and Sellmeier oscillator models are employed to calculate the optical band gap energy, dispersion parameters, and optoelectronic parameters of the loaded (PMMA-PS) thin films. Un-doped PMMA-PS exhibits a high transparency of about 87%. The transmittance decreases dramatically to a vanishing value in the high energy region (250≤λ≤350) nm. Reflectance is found to increase as the concentration of ceria NPs loads increases. Furthermore, refractive index *n* of PMMA-PS exhibits values ranging between 1.76 and 2.13. Interestingly, introducing 7% of CeO_2_ NPs into polymeric matrix leads to a slight increase of *n* to 1.85–2.26. Therefore, PMMA-PS/CeO_2_ nanocomposite could be used for high reflective coatings and candidates for strong optical confinement applications. The optical band gap obtained of PMMA-PS copolymers thin films is ≈4.03 indicating that it is an insulating dielectric material. Introducing CeO_2_ nanoparticles into the copolymers matrix decreases the optical band gap and thus it is possible to engineer the optical properties of this novel material. To elucidate a deeper understanding of the vibrational modes of PMMA-PS/CeO_2_ nanocomposite thin films, we carry out FTIR measurements. We identify and interpret all vibrational bands associated with formation, rotation, and twisting of different bonds involved in the investigated polymerized thin film. Evidently, major changes are observed in width and intensity of the vibrational bands of PMMA-PS upon merging of CeO_2_ NPs in copolymers matrix. In addition, TGA and DTG studies demonstrate that introducing higher concentrations of CeO_2_ NPs into PMMA-PS nanocomposite enhances thermal stability significantly. Surface morphology of PMMA-PS/CeO_2_ NPs at 20 μm magnification shows that PMMA-PS exhibit an amorphous nature with a smooth surface. The SEM images show homogenous dispersion of CeO_2_ NPs on the surface of the PMMA-PS thin films. Having obtained an interesting result on exploiting and understanding the physical mechanisms behind tuning the optical parameters of the polymer-inorganic filler nanocomposites, we are motivated to investigate the effect of changing the types of inorganic fillers, as well as their compositional content on tuning optical parameters of different polymer nanocomposites. Such future investigations are predicted to yield organic-inorganic systems of prime importance for the fabrication of state-of-art optoelectronic multifunctional devices. Furthermore, we are planning to introduce transition metal oxides into different polymeric matrices to examine the possibility of inducing strong magnetic properties to fabricate onto magnetic devices.

Our detailed and comprehensive investigations of the optical, morphological, lattice dynamical, and thermal properties of PMMA-PS/CeO_2_ NPs nanocomposite thin films reveal that they could be utilized in manufacturing realistic-scaled smart multifunctional devices.

## Figures and Tables

**Figure 1 polymers-13-01158-f001:**
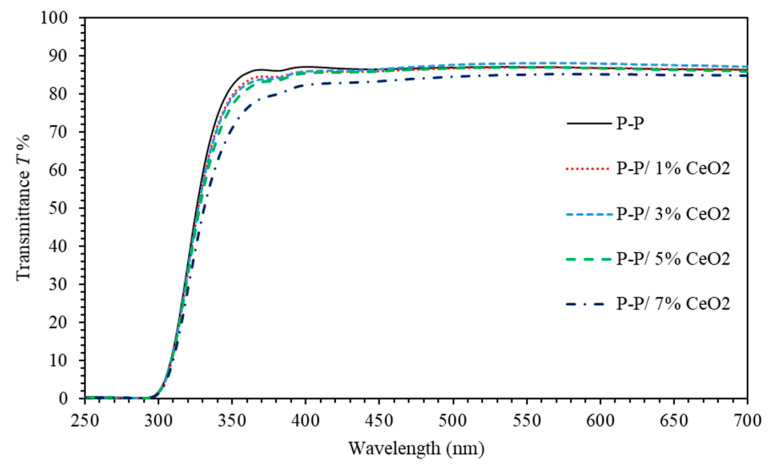
The transmittance of (PMMA-PS)/CeO_2_ nanocomposite thin films incorporated with different CeO_2_ NPs concentrations.

**Figure 2 polymers-13-01158-f002:**
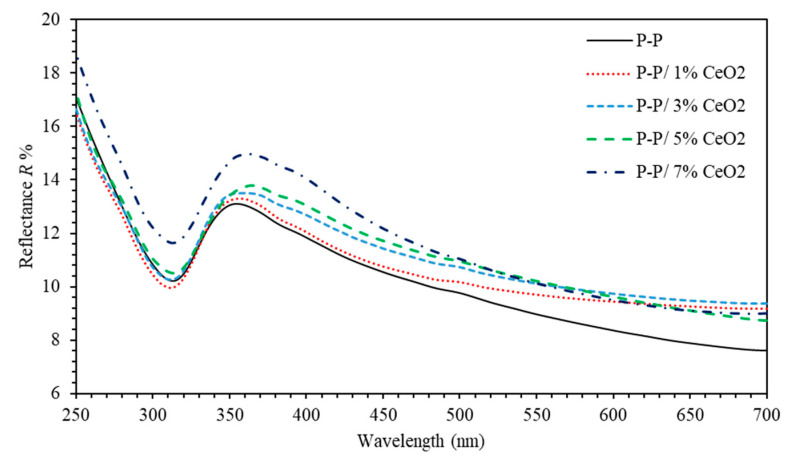
The reflectance of (PMMA-PS)/CeO_2_ nanocomposite thin films with various CeO_2_ NPs concentrations.

**Figure 3 polymers-13-01158-f003:**
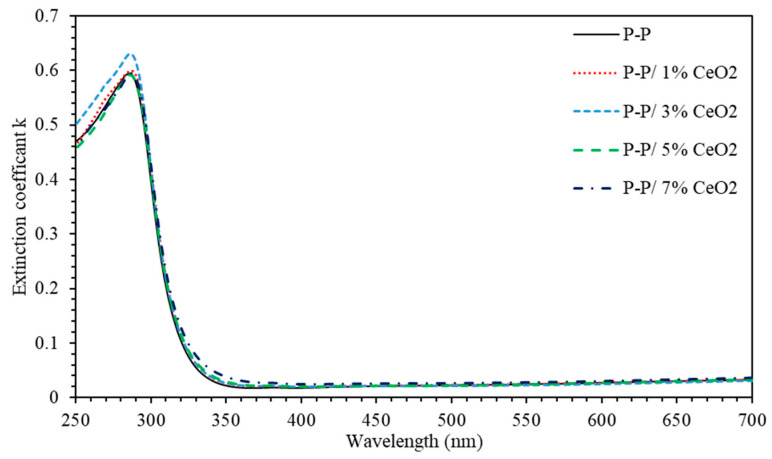
Extinction coefficient *k* of (PMMA-PS)/CeO_2_ nanocomposite thin films with various CeO_2_ NPs concentrations.

**Figure 4 polymers-13-01158-f004:**
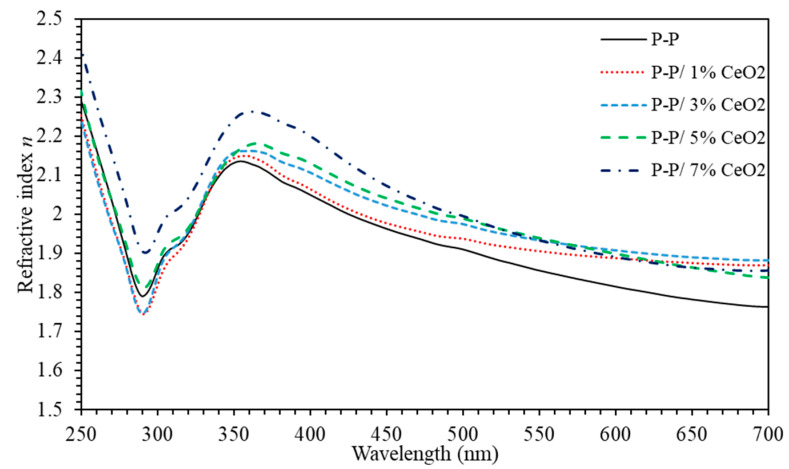
The refractive index (*n*) of (PMMA-PS)/CeO_2_ nanocomposite thin films with various CeO_2_ NPs concentrations.

**Figure 5 polymers-13-01158-f005:**
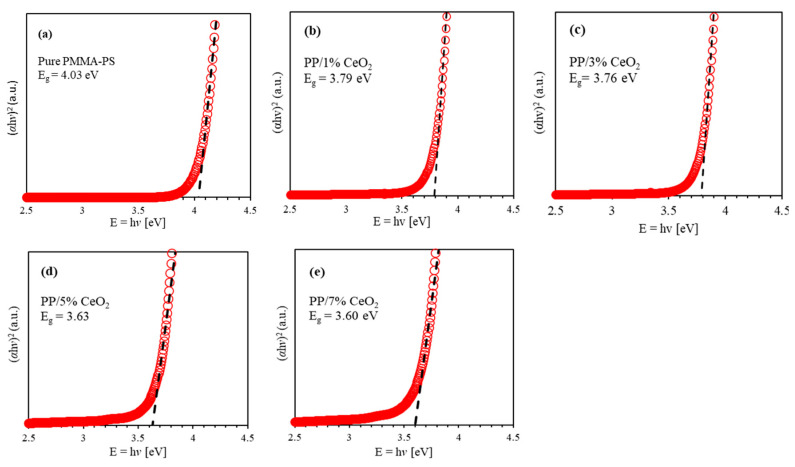
The Tauc plot for energy band gap *E_g_* of (PMMA-PS)/CeO_2_ nanocomposite thin films containing various CeO_2_ NPs concentrations. (**a**) Band gap of Pure PMMA-PS; (**b**) Band gap of PMMA-PS/1% CeO_2_; (**c**) Band gap of PMMA-PS/3% CeO_2_; (**d**) Band gap of PMMA-PS/5% CeO_2_; (**e**) Band gap of PMMA-PS/5% CeO_2_.

**Figure 6 polymers-13-01158-f006:**
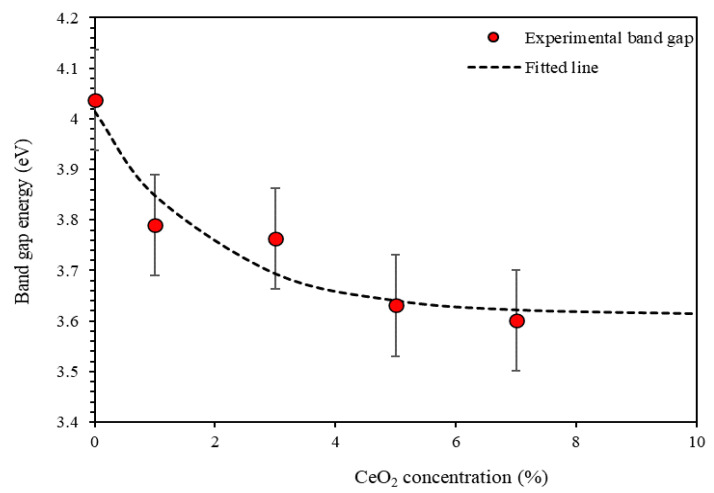
The energy band gap *E_g_* of PMMA-PS as a function of CeO_2_ concentration.

**Figure 7 polymers-13-01158-f007:**
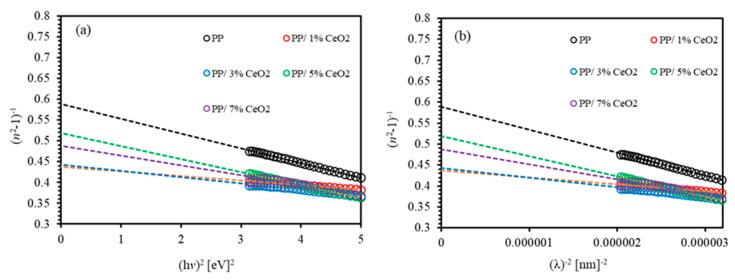
(**a**) The (*n*^2^ − 1)^−1^ versus (*hv*)^2^; (**b**) The (*n*^2^ − 1)^−1^ versus (*λ*)^−2^ of (PMMA-PS)/CeO_2_ nanocomposite thin films with various CeO_2_ NPs concentrations.

**Figure 8 polymers-13-01158-f008:**
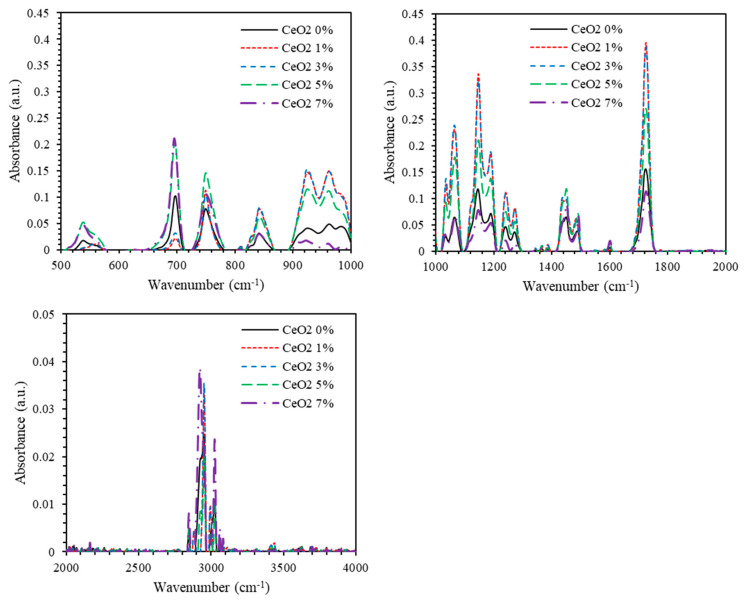
The FTIR spectra of PMMA-PS, and PMMA-PS doped by CeO_2_ NPs.

**Figure 9 polymers-13-01158-f009:**
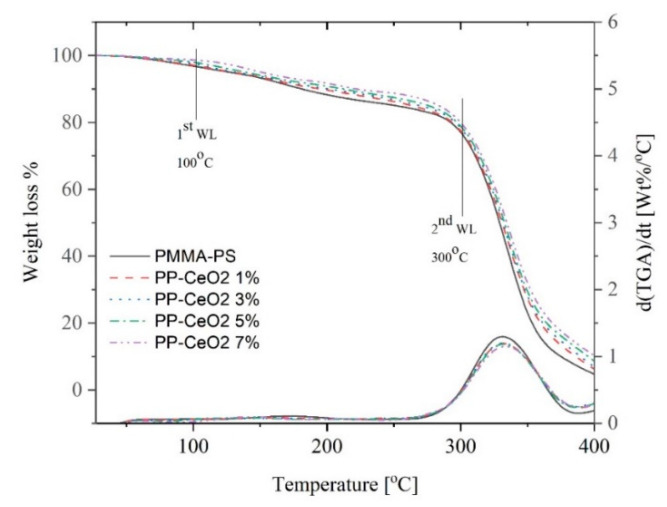
The weight loss and 1st derivative of weight loss of PMMA-PS incorporated with different concentrations of CeO_2_ NPs as a function of temperature.

**Figure 10 polymers-13-01158-f010:**
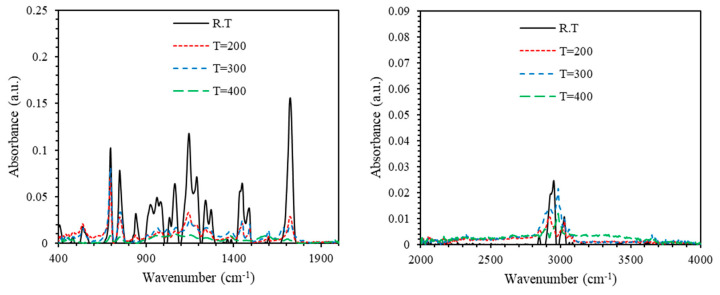
The absorbance of PMMA-PS processed at different annealing temperatures.

**Figure 11 polymers-13-01158-f011:**
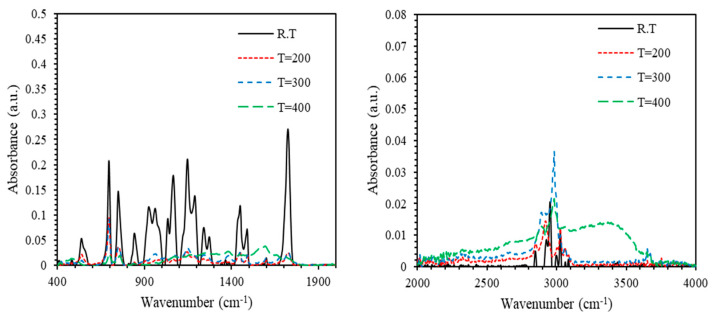
The absorbance of PMMA-PS/5% CeO_2_ processed at different annealing temperatures.

**Figure 12 polymers-13-01158-f012:**
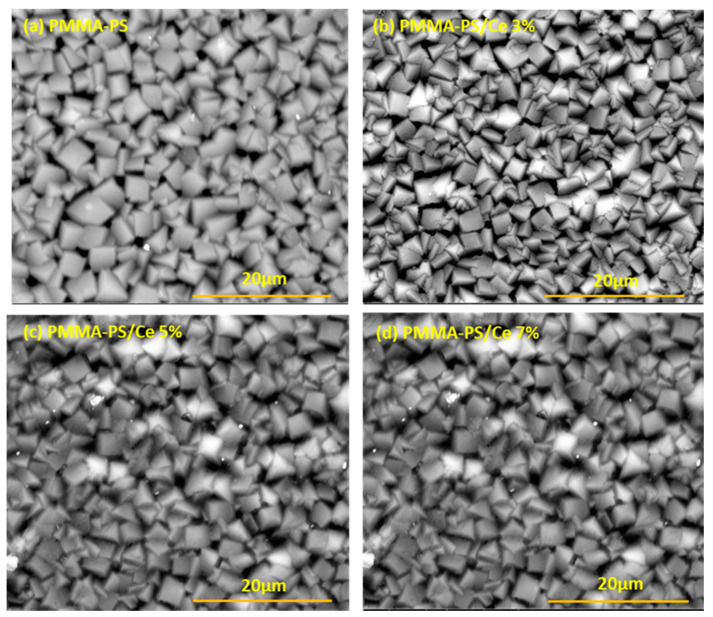
SEM (scanning electron microscope) cross sections of chemically developed images of thin-films of (**a**) PMMA-PS (polymethyl methacrylate-polystyrene) block copolymers, (**b**) PMMA-PS/CeO_2_ 3%, (**c**) PMMA-PS/CeO_2_ 5%, and (**d**) PMMA-PS/CeO_2_ 7%.

**Table 1 polymers-13-01158-t001:** Some optical parameters for PMMA-PS and PMMA-PS incorporation with CeO_2_ NPs.

Parameter	P-P	P-P/1% CeO_2_ NPs	P-P/3% CeO_2_ NPs	P-P/5% CeO_2_ NPs	P-P/7% CeO_2_ NPs
Dispersion energy ***E_d_*** (eV)	6.893	14.484	12.268	7.832	9.288
Effective single oscillator ***E*_0_** (eV)	4.063	6.333	5.433	4.065	4.542
Zero-frequency refractive index ***n*_0_**	1.642	1.812	1.804	1.710	1.744
Zero-frequency dielectric constant ***ε*_0_**	2.696	3.286	3.257	2.926	3.044
Optical oscillator strength ***f*** (eV)^2^	28.011	91.743	66.666	31.847	42.194
Optical moments ***M*_−1_**	1.696	2.286	2.257	1.926	2.044
Optical moments ***M*_−3_** (eV^−2^)	0.102	0.056	0.076	0.116	0.099
Oscillator length strength ***S*_0_ × 10^−5^**	1.804	5.728	4.139	2.119	2.523
Average oscillator wavelength ***λ*_0_**	306.200	199.389	232.683	302.325	282.096
Urbach energy ***E_U_*** (meV)	182.495	186.216	187.360	194.476	207.675

**Table 2 polymers-13-01158-t002:** The peak positions of all vibrational bands of PMMA-PS, and PMMA-PS doped with CeO_2_ NPs.

Vibrational Band	PMMA-PS	PMMA-PS/CeO_2_ 1%	PMMA-PS/CeO_2_ 3%	PMMA-PS/CeO_2_ 5%	PMMA-PS/CeO_2_ 7%
Ce–O	--	541.42	541.42	541.42	541.42
C–H bending	704.03	706.03	693.74	697.86	697.86
	753.44	7.45.20	753.44	751.38	753.44
	841.96	844.02	844.02	844.02	841.96
	965.47	963.41	963.41	963.41	963.41
C–O stretching	1035.46	1037.52	1037.52	1037.52	1035.46
	1068.40	1068.40	1066.34	1068.40	1070.46
	1148.68	1148.68	1148.68	1148.68	1150.47
	1187.79	1191.91	1191.91	1191.91	1191.91
	1251.61	1243.38	1243.38	1245.44	1245.44
	1280.43	1274.26	1274.26	1276.32	1276.32
–CH_3_ bending	1371.01	1366.89	1366.89	1366.89	1368.95
	1389.54	1385.42	1387.48	1385.42	1385.42
–CH_2_ bending	1449.24	1451.29	1447.18	1449.24	1451.29
C=C	1484.23	1486.29	1486.29	1490.41	1492.47
C=O	1601.57	1599.51	1603.63	1603.63	1601.57
	1725.03	1725.08	1725.08	1725.08	1725.08
C–H stretching	2800–3200	2800–3200	2800–3200	2800–3200	2800–3200

## Data Availability

Data availability upon request.

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
