# Peer review of "Synthesis, Optical, Chemical and Thermal Characterizations of PMMA-PS/CeO2 Nanoparticles Thin Film"

_polymers, 2021, doi:10.3390/polym13071158_

Round 1
Reviewer 1 Report
The authors describe the properties of thin films from Poly(methyl methacrylate)-Poly(styrene) copolymer doped with various concentrations of ceria and deposited on glass substrates. It could be an interesting work but there are too many errors mainly in the interpretation of the various data. The description is rather based on the wishful thinking of the authors and not on the presented data. The following are indicative:
- Page 1 line 38: A citation on the high refractive index n of inorganic materials please.
- Page 1 lines 53-55 “Intensive search for smart materials indicates that some polymers have high refractive index as they must be manufactured from monomers.” I do not understand please clarify.
- Page 1 lines 71-72 “Polystyrene is less expensive 71 and has a higher refractive index of 1.59 at a wavelength of 632 nm [4]. “ Higher than what?
- The characterization of the composite membranes leads to the unbiased conclusion that the incorporation of CeO2 nanoparticles enhances their overall properties and the profound question why the authors didn’t try denser concentration in CeO2
- Page 7 Lines 193-195. “Careful examination of the values of ?? and ?? indicate that both increase as CeO2 NPs content introduced into copolymers matrix is increased.” I examined ?? and ?? as carefully as I could but 9.288 and 4.542 did not get bigger than 14.484 and 6.333 for ?? and ?? respectively… Please correct the error.
- Page 7 Lines 210-212 “As demonstrated by Table 1, the values of optical moments ?−1 increases and ?−3 deceases as the concentration of CeO2 NPs is increased suggesting that polarization is reduced [58]. Same here. There is a certain tendency but only up to 5% CeO2 NPs perhaps there is a plateau in higher concentrations another good reason to try denser concentrations in CeO2
- Page 8 Lines 221-223 “The calculated values of ?0 and ?0 are tabulated in Table 2. It reveals that ?0 increases and ?0 decreases as the concentration of CeO2 NPs incorporated into PMMA-PS matrix increases.” First of all, there are in Table 1. Second, they are vice versa again ?0 decreases and ?0 increases and these tendencies hold again up to 5% in CeO2.
- Page 8 line 228 “versus”
- Page 9 Line 245 “The vibrational bands located in the 1000–1300 cm–1 range are assigned to C–O stretching.” There are too many IR bands (6 from Table 2) and only two possible vibrations: symmetric and asymmetric. Further explanations on this assignment are needed.
- Page 11 269-270 “In the second region, mainly observed in the (300-400)oC range with a weight loss ranging (90-20) %” 90-80% I assume.
- Page 11 Lines 270-271 “Such a large weight loss is a clear indication of intermolecular/intramolecular bonding and chemical stability.” This is quite shocking! Such a large weight loss is clear indication of thermal decomposition.
- Page 11 Lines 272-274 “This maximum rate is shifted toward high temperatures as the concentration of CeO2 NPs inserted into PMMA-PS matrix is increased.” It is not! All the first derivative curves peak at the same temperature.
- Page 11 Lines 274-275 Therefore, introducing higher concentrations of CeO2 NPs into PMMA-PS nanocomposite enhances thermal stability of PMMA-PS significantly. The conclusion about the enhancement of thermal stability is completely wrong. The small difference of the Tg curves is due to the higher percentage of CeO2 that does not decompose.
- Page 13 Lines 302-304 Figure 12(a) shows that the nanocomposite thin films of PMMA-PS exhibit an amorphous nature with a smooth surface. I see organized textures and little difference between doped and undoped films.
Author Response
Dear Reviewer 1,
Please find attached the replies to your remarks and suggestions.
Regards,
Prof. Ahmad Alsaad

Reviewer 2 Report
This paper reports an investigation about the synthesis and characterization of PMMA-PS/CeO2 nanoparticles films. Authors also evaluated the optical, chemical and thermal properties of these films. The paper includes interesting results with suitable data analysis. This manuscript may be recommended for publication in Polymers after major revision indicated below.
INTRODUCTION
- “Inorganic” should not be in Capital Letter (Line 37).
- “However, previous studies indicate several drawbacks and insufficient capability of inorganic nanoparticles to serve a variety of modern device applications” – Include references to support this statement.
- Include the main novelty of the study.
MATERIALS AND METHODS
- The materials and methods section (named as “Experimental Details and Techniques”) should be highly improved describing properly the different techniques carried out.
- Did authors perform a statistical analysis of the results obtained? If so, it should be included as a section in the Materials and Methods. If not, authors should carry out one.
RESULTS AND DISCUSSION
- From line 133 to 138, authors described the evolution of the extinction coefficient. However, they did not include a possible reason of this evolution. Please improve the discussion of the results.
CONCLUSIONS
- Mention possible further studies related to this work.
REFERENCES
- Include more references of the journal
- Revise the format of the references and rewrite them following the journal’s Guide for Authors.

Author Response
Dear Reviewer 2,
Please find attached the replies to your comments and suggestions.
Regards,
Prof. Ahmad Alsaad

Reviewer 3 Report
The manuscript entitled " Synthesis, Optical, Chemical and Thermal Characterizations of PMMA-PS/CeO2 Nanoparticles Thin Film " is interesting.
I would suggest the author consider the following items to improve their work:
- Clearly separate Experimental Details and Techniques from Result and Discussion part. There is no need to put some technical details into Discussion part.
- As the Ce NPs are introduced in PMMA-PS thin films, the absorption edge shifts towards lower energy and as a result, a substantial reduction in the band gap energy is noticed. This red shift can be attributed to what? Does that lead to a drastic change in the band structure of the material? Also, please put the Figure 7 in section 3.1.3. and complete presentation
- Does the considerable increase in the peak's intensities of the whole FTIR spectra could be attributed to the intermolecular bonding between PMMA-PS matrix and Ce NPs?
- How the SEM image confirms the presence of Ce NPs? Did you perform identification by EDS analysis?
Cerium oxide nanoparticles purchased from (Sigma Aldrich) of size (25-50) nm were added to copolymers matrix, but CeO2 NPs observed by SEM, were within dimension the average size between (100–500) nm in diameter. How do you explain this?
Also, technically arrange the Figure 12, so that the images go from a to d, one after the other.
I recommend acceptance for publication after minor modifications.
Author Response
Dear Reviewer 3,
Please find attached the replies to your comments and suggestions.
Regards,
Prof. Ahmad Alsaad

Round 2
Reviewer 1 Report
In some cases, the authors complied with the reviewer’s comments and corrected the mistakes as for example in 1-3, 5,8.
In some cases, they did not: Page 11 Lines 329-332. “In the second region, mainly observed between 300 oC and 400 oC, the weight loss decreases from 90% to 20%. Such a large weight loss is clear indication of thermal decomposition suggesting that doped polymeric thin films exhibit low chemical stability at high temperature.” If the weight loss decreases it can’t be large! Actually, the mistake begins at the y axis of Figure 9. The diagram does not depict weigh loss. It is weight as percentage of the initial weight before thermal treatment (Weight as percentage of W0 (%). Weight loss (%) is 100-Y. Thus, weight loss INCREASES from 10% to 80% (actually 90%) and now the paragraph makes sense.
In some cases, they did half the job. They corrected their wrong statement about the enhancement of the thermal stability of PMMA-PS thin films by the CeO2 nanoparticles in the Results and Discussion Section (which was a crucial issue for the initial rejection) but not in the conclusions section. Page 14 lines 393-5.
In some others they managed to produce new ones see for example Page 9 Lines 270-2 “The calculated values of ?0 and ?0 are tabulated in Table 1. It reveals that ?0 and ?0 of PMMA-PS are found to be 306.200 x10-5 and 306.200 nm, respectively.” instead of 1.804x10-5.
The assignment of the IR at the 1300-1000 cm-1 bands and the justification are both wrong and both without any reference. In fact, there are no references in the IR section. One of the bands at 1060 cm-1 is C-C stretching of the PMMA backbone (Irving Lipschitz (1982) The Vibrational Spectrum of Poly(Methyl Methacrylate): A Review, Polymer-Plastics Technology and Engineering, 19:1, 53-106, DOI: 10.1080/03602558208067727; STOIL DIRLIKOV and JACK L. KOENIG Infrared Spectra of Poly(Methyl Methacrylate) Labeled with Oxygen-18 APPLIED SPECTROSCOPY 1979, Volume 33, Number 6, 551-555) and most probably the out of phase in plane C-H bending of C-H bonds of the aromatic ring of polystyrene (Nyquist, R. A., Putzig, C. L., Leugers, M. A., McLachlan, R. D., & Thill, B. (1992). Comparison of the Vibrational Spectra and Assignments for α-Syndiotactic, β-Syndiotactic, Isotactic, and Atactic Polystyrene and Toluene. Applied Spectroscopy, 46(6), 981–987. doi:10.1366/0003702924124321). For the assignment of the other bands, I suggest the above 3 manuscripts and references within.
All the above issues are not critical for a second rejection. There was though a (strong)suggestion to the authors to extend their experiments to higher contents of CeO2 in order to determine the optimum content. Their answer: "In this work, we focused on diluted concentrations of CeO2 to elucidate its effect on the properties of PMMA-PS polymeric thin films. In future works, we will investigate the effect of denser concentrations of CeO2 on the chemical and physical properties of PMMA-PS/CeO2 Nanoparticles Thin Films.” Cutting a scientific work into pieces in not acceptable, at least not by me.
Reviewer 2 Report
Authors carried out all the changes suggested during the previous review process. In this sense, this manuscript is recommended for publication after the verification that the "References Section" is written following the Authors' Guidelines.